# The Impact of Melanoma Imaging Biomarker Cues on Detection Sensitivity and Specificity in Melanoma versus Clinically Atypical Nevi

**DOI:** 10.3390/cancers16173077

**Published:** 2024-09-04

**Authors:** Rosario Agüero, Kendall L. Buchanan, Cristián Navarrete-Dechent, Ashfaq A. Marghoob, Jennifer A. Stein, Michael S. Landy, Sancy A. Leachman, Kenneth G. Linden, Sandra Garcet, James G. Krueger, Daniel S. Gareau

**Affiliations:** 1Department of Dermatology, Escuela de Medicina, Pontificia Universidad Católica de Chile, Santiago 8330024, Chile; 2Department of Dermatology, Medical College of Georgia at Augusta University, Augusta, GA 30904, USA; 3Melanoma and Skin Cancer Unit, Escuela de Medicina, Pontificia Universidad Católica de Chile, Santiago 8330024, Chile; ctnavarr@uc.cl; 4Memorial Sloan Kettering Skin Cancer Center, New York, NY 10022, USA; marghooa@mskcc.org; 5Ronald O. Perelman Department of Dermatology, New York University Grossman School of Medicine, New York, NY 10016, USA; jennifer.stein@nyulangone.org; 6Department of Psychology, New York University, New York, NY 10003, USA; landy@nyu.edu; 7Center for Neural Science, New York University, New York, NY 10003, USA; 8Dermatology Department, Oregon Health & Science University, Portland, OR 97239, USA; leachmas@ohsu.edu; 9Dermatology Department, University of California Irvine, Irvine, CA 92868, USA; 10Chao Family Comprehensive Cancer Center, University of California Irvine, Irvine, CA 92868, USA; 11Laboratory for Investigative Dermatology, Rockefeller University, New York, NY 10065, USA; sandra.garcet@rockefeller.edu (S.G.); james.krueger@rockefeller.edu (J.G.K.)

**Keywords:** dermoscopy, artificial intelligence, melanoma, imaging biomarkers

## Abstract

**Simple Summary:**

Early detection of melanoma and differentiation from benign nevi can be challenging even for the most experienced dermatologists. To improve melanoma detection, artificial intelligence algorithms incorporating dermoscopy have been developed, but lack transparency and therefore have limited training value for healthcare providers. To address this, an automated approach utilizing imaging biomarker cues (IBCs), logical features extracted from images that mimic expert dermatologists’ dermoscopic pattern recognition skills, was developed. This study excluded deep learning approaches to which IBCs are complementary or alternative. Ten participants assessed 78 dermoscopic images (39 melanomas and 39 nevi) first without IBCs and then with IBCs. Using IBCs significantly improved diagnostic accuracy: sensitivity increased significantly from 73.69% to 81.57% (*p* = 0.0051) and specificity increased from 60.50% to 67.25% (*p* = 0.059). These results indicate that incorporating IBCs can significantly enhance melanoma diagnosis, with potential implications for improved screening practices. Further research is needed to confirm these findings across a variety of healthcare providers.

**Abstract:**

Incorporation of dermoscopy and artificial intelligence (AI) is improving healthcare professionals’ ability to diagnose melanoma earlier, but these algorithms often suffer from a “black box” issue, where decision-making processes are not transparent, limiting their utility for training healthcare providers. To address this, an automated approach for generating melanoma imaging biomarker cues (IBCs), which mimics the screening cues used by expert dermoscopists, was developed. This study created a one-minute learning environment where dermatologists adopted a sensory cue integration algorithm to combine a single IBC with a risk score built on many IBCs, then immediately tested their performance in differentiating melanoma from benign nevi. Ten participants evaluated 78 dermoscopic images, comprised of 39 melanomas and 39 nevi, first without IBCs and then with IBCs. Participants classified each image as melanoma or nevus in both experimental conditions, enabling direct comparative analysis through paired data. With IBCs, average sensitivity improved significantly from 73.69% to 81.57% (*p* = 0.0051), and the average specificity improved from 60.50% to 67.25% (*p* = 0.059) for the diagnosis of melanoma. The index of discriminability (*d*′) increased significantly by 0.47 (*p* = 0.002). Therefore, the incorporation of IBCs can significantly improve physicians’ sensitivity in melanoma diagnosis. While more research is needed to validate this approach across other healthcare providers, its use may positively impact melanoma screening practices.

## 1. Introduction

Cutaneous invasive melanoma in the form of a distinct, irregular, variegated black or brown patch or ulcerated plaque is clinically identifiable by most healthcare professionals, and its recognition prompts heightened diagnostic evaluation. In these straightforward presentations, the clinician often relies on foundational melanoma screening guidelines such as the ABCD or ABCDE criteria to critically assess and guide further workup [1,2]. However, for non-invasive melanomas, like melanoma in situ and lentigo maligna, or for early invasive melanomas without features of disorganization or those with smaller diameters, providers may not be prompted to use the ABCDE criteria or other screening methods, which may result in decreased melanoma detection [3,4].

Early detection of melanoma is vital to reducing the morbidity and mortality associated with late-stage disease. The American Cancer Society estimates that for 2024, approximately 100,640 new melanomas will be diagnosed (about 59,170 in men and 41,470 in women) and 8290 people will die of melanoma (about 5430 men and 2860 women) [5]. The annual cost burden exceeds 1 billion USD [6], including the high cost of treating late-stage disease. Earlier detection (diagnosis of Stage 1A) virtually eliminates mortality and decreases the cost burden by enabling cure with excision alone [7].

There are several significant needs relating to early melanoma detection, including better tools to identify subtle presentations, melanoma subtype-specific algorithms, and the recognition of melanoma subtype-specific at-risk phenotypes and genotypes [8]. Additionally, access to dermatology care is often delayed [9], and when patients are evaluated during a face-to-face visit, the decision to biopsy any particular melanocytic lesion has its challenges. To address these challenges, numerous tools and technologies, such as dermoscopy, have been developed to aid providers in the decision-making process. Still, in dermatologic specialty practice, only 3–25% [10] or a mean of 10% [11] of excised suspicious lesions are diagnosed as melanoma. While only approximately 48% of United States (US) dermatologists are reported to utilize dermoscopy [12], it has shown to significantly improve clinician’s diagnosis of melanoma compared to a naked-eye exam alone [13,14,15], with an increase in sensitivity ranging from 10 to 27% [16]. Dermoscopy non-users cite that barriers to implementation include cost and lack of reimbursement, which encompasses both time and dedicated training [17,18]. Even for dermatology residents, education in dermoscopy is not yet standardized, as there is considerable variation in the amount of didactic dermoscopy lectures for individual residency programs, a lack of access to pigmented lesion specialists, and significant variation in resident dermoscopy resources [18,19]. Furthermore, even for those with experience, there remains an unmet need to help differentiate between benign and malignant lesions, especially clinically atypical nevi from melanoma [20].

In recent years, artificial intelligence (AI) algorithms and devices have been developed to help physicians diagnose melanoma and other malignant skin neoplasms [21,22,23]. However, the “black box” nature of these AI systems does not currently allow physicians to identify new clinical or dermoscopic lesional characteristics that could be beneficial in future clinical assessments. While experts in dermoscopy have a high diagnostic accuracy (in terms of sensitivity and specificity) in identifying malignant lesions [14,16,24,25], there is a continued need for innovative diagnostic approaches to support a wider range of healthcare practitioners in melanoma detection [26].

This problem underscores the importance of further technological developments in melanoma detection. In the last several years, an AI approach for generating quantitative imaging analysis metrics, known as imaging biomarker cues (IBCs), has been demonstrated [27,28]. Melanoma IBCs are quantitative dermoscopy image features that predict melanoma diagnosis. IBCs translate the right amount of appropriate information from machine vision to the clinician, even from invisible device imaging wavelengths and associated features. This method utilized a set of 13 machine-learning algorithms to calculate an overall malignancy risk score, demonstrating 98% sensitivity and 36% specificity in melanoma detection. However, its efficacy in aiding dermatologists in differentiating between melanoma and atypical nevi has not been tested.

In this study, we aim to evaluate the accuracy of dermatologist screening in identifying melanoma versus clinically atypical nevi using IBCs within a human-subject-research setting.

To assess the accuracy of dermatologists’ screening with and without the inclusion of IBCs, ten dermatologists with varying levels of expertise were recruited. Their performance in distinguishing between dermoscopic images of clinically atypical nevi and melanomas was evaluated both with and without IBCs.

## 2. Materials and Methods

### 2.1. Participant Background and Recruitment

The previous study developed IBCs [27,28,29] and study data [28] and was approved by the Rockefeller University Institutional Review Board. For this study, ten participants (seven dermatologists, one dermatology resident, and two dermatopathologists) with experience in dermoscopy were recruited. All participants completed basic dermoscopy training. Their expertise ranged from novice (dermatology resident) to expert (three expert dermoscopists). The group also included two dermatopathologists and four Mohs surgeons. The three expert dermoscopists had a minimum of three years of experience practicing dermoscopy. They either received extensive dermoscopy training through dedicated courses and applicable clinical experience or were fellowship-trained by leaders in the field. They were also heavily involved in dermoscopy education at academic institutions or national conferences and specialized in the evaluation of pigmented lesions.

### 2.2. Participant Training

After participant recruitment, on a video call, a script was read orally to each participant, designed to be straightforward and efficient, simulating the rapid decision-making process in clinical settings (Appendix A). During the training, participants were instructed on how to use IBCs in image evaluation. The training duration for participants was limited to the time taken to deliver the human-subjects-research script, which was approximately one minute.

### 2.3. Dermoscopy Image Acquisition and Presentation

A total of 78 dermoscopic images were used, with an equal distribution of melanomas and nevi (Appendix A). We excluded 30% of the original 112 images that had the fewest pixels to maintain consistency in quality while ensuring a balanced representation of melanoma and nevus cases. The images utilized in this study were acquired from a cohort of 39 primary melanomas and 39 nevi. They were captured using an EpiFlash™ (Canfield Inc., Parsippany-Troy Hills, NJ, USA) dermatoscope attached to a Nikon D80 camera. The resulting alcohol-coupled, non-polarized dermoscopy images (post cropping) contained resolutions ranging from one to five megapixels at 300 dpi and were all obtained in New York at the Memorial Sloan Kettering Cancer Center. 

The dermoscopy images were presented in a randomized order except for the first two images. These were specifically chosen based on their clarity: an obvious nevus and melanoma (Figure 1). Following these two images, each participant reviewed the additional images, first without IBCs and then with the addition of IBCs.

The images were shown to the study participants as described by the sequence in Figure 2E. The protocol included revealing pathological diagnoses (Figure 2C, top right) to approximate the memory recall/bias that can occur during new technology adoption by clinicians. Figure 2A shows the first step, where images were displayed without IBCs; Figure 2B shows the second step, in which images were displayed with IBCs. Participants were asked to classify each image as either melanoma or nevus in both steps. This design enabled the collection of paired data for comparative analysis. In the second step, participants were asked to base their decisions on three factors: (1) the ensemble classifier score (ranging from one for melanoma to zero for nevus); (2) a single IBC; and (3) their clinical instinct or “gestalt” approach. The single IBC, termed the “clock-sweep melanoma radar”, quantified brightness on a clock arm connecting the center of the lesion to the peripheral margin as a function of the angle, to provide a graphical and numerical value for the irregularity in lesion brightness. This was displayed by the clock hands (blue, green, and red) along with the numerical value (Δ/mean). 

### 2.4. Outcome Measures

The primary outcome measures included the accuracy of melanoma diagnosis with and without the use of IBCs. By comparing the classifications made by the participants with the gold-standard pathological diagnoses, the effectiveness of IBCs in improving diagnostic accuracy in melanoma detection was calculated. Statistical significance was assessed using a *t*-test comparing paired proportions to calculate the *p*-value.

## 3. Results

Invasive melanoma is an aggressive form of skin cancer that can present diagnostic challenges, particularly in discrimination from melanocytic nevi [20]. Melanoma may present with many subtle features and its diagnosis relies heavily on the healthcare provider’s ability to discern malignant characteristics within the lesion. There is a critical need for more effective screening techniques, as melanoma mortality often advances with delayed diagnosis. In recent years, AI algorithms have been developed to help physicians diagnose melanoma and other malignant skin neoplasms [21,22,23]. However, the “black box” nature of these systems does not currently allow physicians to identify new clinical or dermoscopic lesional characteristics that could be beneficial in future clinical assessments. Here, we present the accuracy of dermatologist screening in identifying melanomas versus nevi with and without the use of IBCs.

### 3.1. Dermoscopy Image Evaluation Sequence

To evaluate the accuracy of dermatologist screening in identifying melanomas versus nevi using IBCs, the dermoscopy images were presented in a randomized order except for the first two images. As mentioned, these were specifically chosen based on their clarity: an obvious nevus (Figure 1A) and an obvious melanoma (Figure 1B). Following these two images, each participant reviewed 76 additional images, first without IBCs and then with the addition of IBCs. All images were viewed by participants with resolutions ranging from one to five megapixels at 300 dpi.

An example image sequence shown to study participants is provided in Figure 2, as detailed in Section 2. The images were shown to the study participants as described by the sequence in Figure 2E. Figure 2A shows the first step, where images were displayed without IBCs; Figure 2B shows the second step, where images were displayed with IBCs. The pathological diagnosis for each image (Figure 2C, top right) was provided after the first step. Participants classified each image as either melanoma or nevus in both steps. Participant decisions were based on three factors: (1) the ensemble classifier score (ranging from one for melanoma to zero for nevus); (2) a single IBC; and (3) their clinical instinct or “gestalt” approach. The single IBC, termed the “clock-sweep melanoma radar”, quantified brightness on a clock arm connecting the center of the lesion to the peripheral margin as a function of the angle, to provide a graphical and numerical value for the irregularity in lesion brightness. This was displayed by the clock hands (blue, green, and red) along with the numerical value (Δ/mean). 

Figure 3 shows examples of IBCs that participants were shown for nevi (Figure 3A,B) and melanomas (Figure 3C,D). The ensemble classifier score is located at the top left of each image (Risk = x), and the “clock-sweep melanoma radar”, displayed numerically as Δ/mean, is located at the bottom right of each image. 

### 3.2. Participant Performance 

The previously described AI approach for generating quantitative imaging analysis metrics [27,28] yielded the receiver operating characteristic (ROC) for the study participants shown in Figure 4. This method utilized a set of 13 machine-learning algorithms to calculate an overall malignancy risk score with the red point (labeled Eclass Computer Algorithm Alone) specifying a 98% sensitivity and a 36% specificity for melanoma detection, approximating the accuracy of expert lesion evaluation [28]. The performance of each of the ten participants without the use of IBCs (dermoscopy image alone) is indicated by the red dot, whereas the green dot represents the performance of each participant with IBCs. The black line connects the two points for each participant, representing the change in sensitivity and specificity. 

The average sensitivity improved from 73.69% to 81.57% with the use of IBCs (*p* = 0.0026; this and subsequent statistics are matched-sample *t*-tests), while the average specificity increased from 60.50% to 67.25% (*p* = 0.029). The mean increase in sensitivity was 7.87 (SD = 6.79), and the mean increase in specificity was 6.75 (SD = 9.86). The mean index of discriminability (*d*′) increased from 0.94 (SD = 0.38) to 1.41 (SD = 0.35), yielding a significant increase of 0.47 (SD = 0.34, *p* = 0.002). The individual performance of each participant is presented in Table 1.

## 4. Discussion

In this study, we explored the efficacy of IBCs as a tool to aid dermatologists in the diagnosis of melanoma. Ten participants, with various levels of dermoscopy expertise from diverse geographical locations, analyzed dermoscopic images without and with the assistance of IBCs. 

Table 1 shows the sensitivity and specificity of the participants with and without IBCs. Of note, the baseline sensitivity of the dermatology resident was slightly higher than that of the first listed expert dermoscopist. With n = 1, this is not significant but does warrant discussion. One possibility is to assess the improvement of this expert dermoscopist compared to the improvement of the dermatology resident. The expert dermoscopist had an increase in sensitivity of 10.5%, while the resident had an increase in sensitivity of 2.6%. This may be because the expert dermoscopist has more developed pattern analysis skills that are better aligned with the IBC visual language, whereas the dermatology resident, having less experience and lower utilization of these specific skills, did not engage with the visual language as effectively. Additionally, the absence of clinical images and lack of information regarding the typical nevus patterns for each individual and subsequent dermoscopic image represent objective information that was lacking from this study. It may have been challenging for this expert to rely solely on a dermoscopic image for diagnosis when the complete clinical and dermoscopic picture was not available.

We also would like to comment on the performance of one of the Mohs surgeons, listed last in Table 1, who showed a sensitivity decrease of 2.6% with the addition of IBCs. For this participant, the results indicated an unlucky combination of correct and incorrect answers. We observed that this participant began to develop response patterns influenced by the machine’s output. For instance, when the participant gave an incorrect answer when the machine was correct, they became overly reliant on the machine. As a result, the participant placed too much trust in the machine for subsequent evaluations, even when the machine was incorrect. We recognize that neither humans nor machines are perfect, and we conclude that this type of human–machine interaction may involve a form of cognitive manipulation. Therefore, it remains crucial for humans to recognize the importance of clinical judgment when using machine-learning technologies in their practice.

Overall, however, the results showed statistically significant improvements in accuracy, demonstrating that IBCs can be a valuable addition when it comes to enhancing diagnostic precision among dermatologists, even at the expert level. The average sensitivity improved from 73.69% to 81.57% with the use of IBCs, while the average specificity increased from 60.50% to 67.25%. The mean increase in sensitivity was 7.87 (SD = 6.79), and the mean increase in specificity was 6.75 (SD = 9.86). Based on the improvements of the dermatologists, we presume that the use of IBCs may be of significant interest to different clinicians. Mohs surgeons, for example, have a strong background in dermatology but may lack dedicated training in dermoscopy. A strong dermatology background, essential in clinical recognition of malignant lesions, coupled with IBCs, is ideal for this technology. This approach leveraged both clinical intuition and dermoscopic image-based cues, which, even when utilized in the absence of clinical images, appeared to aid participants in distinguishing melanomas from atypical nevi in this study.

In recent years, there has been a growing interest in the use of AI, particularly convolutional neural networks (CNNs), to demonstrate expert-equivalent accuracy in skin cancer diagnosis [22]. However, most of these AI models, including CNNs, are often criticized for their “black box” nature, offering limited insight into the diagnostic process [30,31]. In contrast, IBCs and non-deep (i.e., non-CNN) machine-learning represent a more transparent approach, providing tangible, understandable metrics that aid dermatologists in decision making. Unlike CNNs that operate on complex, often opaque algorithms, IBCs allow for the integration of AI into the clinician’s expertise, potentially enhancing diagnostic accuracy while retaining interpretability. Various classification methods for skin cancers have been developed, each with their own unique ability to evaluate dermoscopic images. Masood and colleagues outline over eight different classification methods, including artificial neural networks, support vector machines, and rule-based classifications, with comparisons of sensitivity and specificity in melanoma detection for over 30 classification systems from the 1990s to 2012. In this comparison, the sensitivity for the various classification systems ranges from as low as 51% to as high as 100%, and the specificity ranges from 70 to 100% [23]. A direct comparison of these methods with the performance of IBCs is difficult, as each of these models has different chosen classifiers, a range of sample sizes, variation in the proportion of melanomas, and marked differences in the number of dermoscopic features evaluated. Recently however, the new 2023 FDA-approved AI-powered medical device utilizing elastic-scattering spectroscopy [21], has demonstrated a 96.67% sensitivity for melanoma and an overall device specificity of 26.22%, with no statistically significant difference found between the device and dermatologist performance. However, this device is primarily marketed for primary care providers and does not utilize dermoscopic images for the device input. 

This study’s findings suggest that IBCs, with clear and interpretable criteria, not only complement but may surpass the diagnostic accuracy of traditional AI methods in melanoma detection [27]. The efficacy of imaging biomarkers is further underscored by their successful application in other medical areas. Magnetic resonance imaging-based radiomics features have been widely adopted for characterizing solid renal neoplasms [32,33]. Similarly, the Paige Prostate, a clinical-grade AI tool, has been instrumental in prostate cancer diagnosis and grading [34]. To the best of our knowledge, there has been little interest in the study of imaging biomarkers for dermatology and skin cancer. The significance of our findings is notable, as dermatologists may effectively enhance their diagnostic accuracy in differentiating nevus from melanoma by incorporating imaging biomarkers. The “clock-sweep melanoma radar”, a novel single imaging biomarker from our previous studies, represents an innovative approach to evaluate irregularity in lesion brightness [27,28]. The use of transparent and understandable metrics like this biomarker may empower physicians to make informed decisions and recognize new dermoscopic characteristics, a process that appears to be lacking in current AI models.

Regarding limitations, this study was conducted with a small cohort of clinicians and with a relatively small number of dermoscopy images, restricting the external validity of our findings. To solidify these results and ensure that they are generalizable across the broader dermatological community, further testing with a larger and more diverse group of dermatologists and images is necessary, along with testing in a clinical setting. Additionally, future work is needed to test IBCs in all Fitzpatrick skin types, ages, and anatomical locations. This study evaluated mostly Fitzpatrick skin types I-III with melanocytic lesions on the trunk and extremities. Another limitation is the applicability of IBCs beyond dermatology specialists, as IBCs could be a valuable tool for other healthcare professionals involved in skin cancer detection and diagnosis. Future studies should aim to validate the use of IBCs across various healthcare professionals, particularly those with some dermoscopy training (e.g., primary care doctors), which could significantly expand the utility of this tool in clinical practice. Finally, the increase in specificity, although positive, reached only borderline statistical significance. However, this outcome should not overshadow the overall positive trends observed in our study, which demonstrated a robust and statistically consistent improvement in sensitivity. 

## 5. Conclusions

Our research demonstrated that the incorporation of IBCs significantly enhanced diagnostic sensitivity in melanoma identification. This study could pave the way for future research to validate the effectiveness of IBCs among other healthcare professionals in broader clinical settings. Further studies could potentially extend vision into the invisible wavelengths of light (i.e., ultraviolet and infrared), as IBCs with displayable features can be created from images acquired at imperceptible wavelengths [29]. We believe that the integration of IBCs into dermoscopic image interpretation has the potential to improve the accuracy of melanoma detection and subsequently improve patient outcomes.

## Figures and Tables

**Figure 1 cancers-16-03077-f001:**
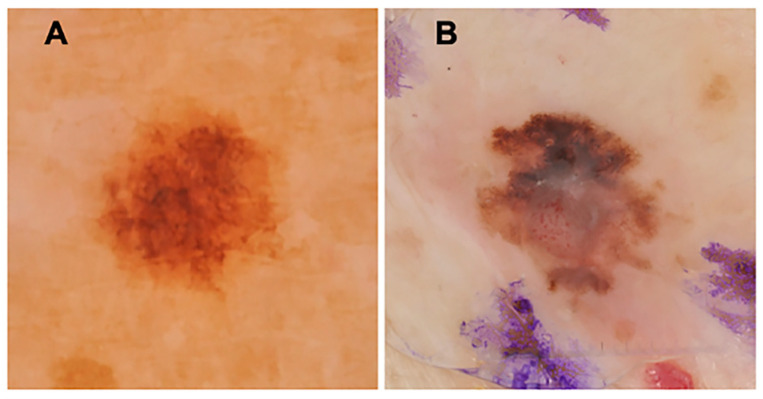
The first two images viewed by study participants due to their simplicity and clarity. (**A**) melanocytic nevus and (**B**) melanoma. The resolution for this figure is 3.2 megapixels, 300 dpi.

**Figure 2 cancers-16-03077-f002:**
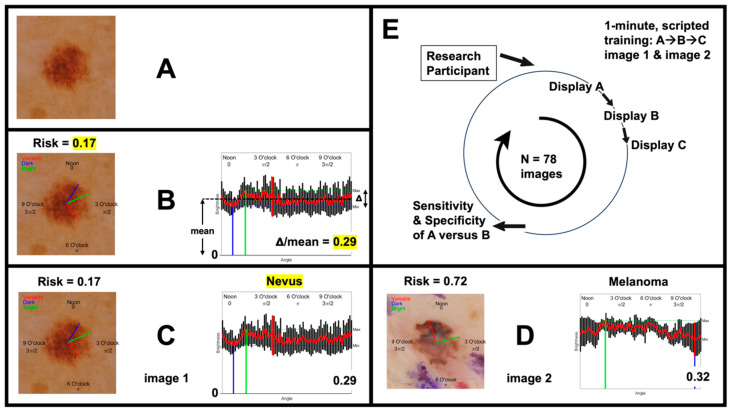
The human-subject-research procedure for the dermatologists first showed the raw dermoscopy image (**A**) with the question, “is this a melanoma or nevus?” After the binary diagnosis, IBCs were shown (**B**) with the same question repeated, to determine the impact of IBCs, if any, on the binary diagnosis through a second binary diagnosis, recorded before the third image display (**C**), which contained the pathology diagnosis (ex. nevus). An example risk score and pathology diagnosis for a melanoma is provided in (**D**). The images were shown to the study participants as described in (**E**). The two numerical risk scores were (1) the ensemble classifier (Eclass) risk score (shown as Risk = x), between zero for nevus and one for melanoma—for this nevus, the risk is 0.17, based on many IBCs and multiple consumed non-deep machine-learning algorithms [1]—and (2) the single IBC on the bottom right (e.g., 0.29), which is the “clock-sweep melanoma radar”, measuring the irregularity in lesion brightness as a function of the angle by quantifying the range in brightness (Δ) divided by the mean brightness. The visual translation from cartesian to polar analysis is shown on an imagined clock face superimposed on the lesion, starting at noon and going clockwise. In (**B**), it reaches the darkest angle (blue line), the brightest angle (green line), then the angle of max brightness variation (red line) before 6 o’clock (“π” radians), pointing directly down, or at the 6 o’clock mark on the clock face.

**Figure 3 cancers-16-03077-f003:**
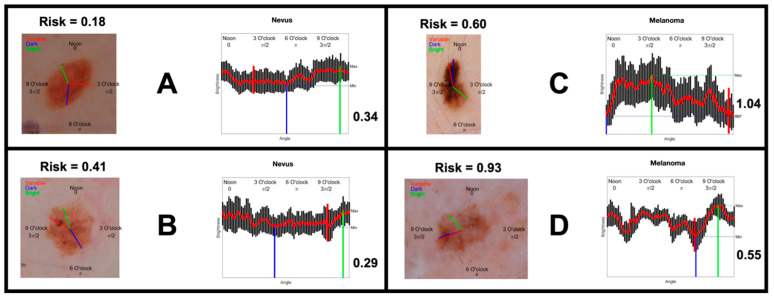
Imaging biomarker cue examples for nevi (**A**,**B**) and melanomas (**C**,**D**).

**Figure 4 cancers-16-03077-f004:**
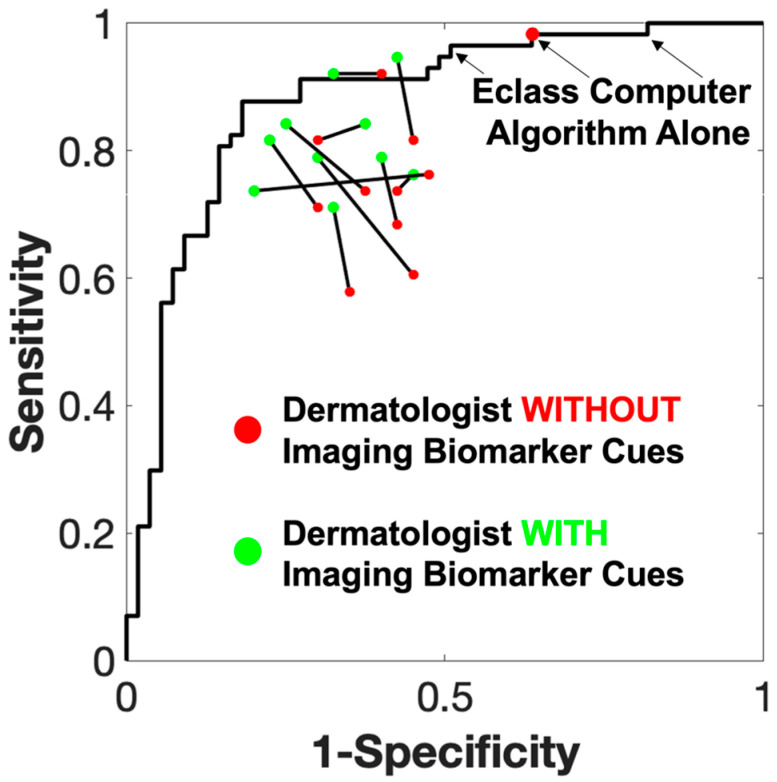
Sensitivity and specificity of dermatologists when provided with imaging biomarker cues during melanoma diagnosis. The ROC curve is for the Eclass algorithm alone, derived from a previous publication [28]. Most dermatologists improved in both sensitivity and specificity, as can be seen in the gain from the red point, which measured their accuracy without IBCs, to the green point, which is their accuracy with IBCs.

**Table 1 cancers-16-03077-t001:** Accuracy of each of the participants studied without and with IBCs during sensory cue integration and melanoma diagnosis.

	Sensitivity without IBCs	Specificity without IBCs	Sensitivity with IBCs	Specificity with IBCs	Sensitivity Increase	Specificity Increase	*d*′ without	*d*′ with	*d*′ Increase
Expert Dermoscopist	71.1	70	81.6	77.5	10.5	7.5	1.08	1.66	0.57
Dermatopathologist	81.6	55	94.7	57.5	13.1	2.5	1.03	1.81	0.78
Expert Dermoscopist	92.1	60	92.1	67.5	0	7.5	1.67	1.87	0.2
Mohs Surgeon	68.4	57.5	78.9	60	10.5	2.5	0.67	1.06	0.39
Dermatology Resident	73.7	57.5	76.3	55	2.6	−2.5	0.82	0.84	0.02
Dermatopathologist	81.6	70	84.2	62.5	2.5	−7.5	1.42	1.32	−0.1
Mohs Surgeon	57.9	65	71.1	67.5	13.2	2.5	0.58	1.01	0.43
Mohs Surgeon	60.5	55	78.9	70	18.4	15	0.39	1.33	0.94
Expert Dermoscopist	73.7	62.5	84.2	75	10.5	12.5	0.95	1.68	0.72
Mohs Surgeon	76.3	52.5	73.7	80	−2.6	27.5	0.78	1.48	0.7
Mean	73.69	60.50	81.57	67.25	7.87	6.75	0.94	1.41	0.47

## Data Availability

The original contributions presented in the study are included in the article and Appendix A; further inquiries should be directed to the corresponding author. The Eclass data were generated from prior studies [1,2].

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
