# Peer review of "The Impact of Melanoma Imaging Biomarker Cues on Detection Sensitivity and Specificity in Melanoma versus Clinically Atypical Nevi"

_cancers, 2024, doi:10.3390/cancers16173077_

Round 1
Reviewer 1 Report
Comments and Suggestions for Authors
The work titled as "The Role of Digital Imaging Biomarkers for Melanoma Detection Among Dermatologists: A reader study" was presented with the goal to better identify the biomarkers in Melanoma. The overall goal is promising but the manuscript needs to be improved. I have some comments which need to be implemented.
1. Title must be updated so that it must look like research article and must represent the overall work presented here. My suggested title: "A deep learning based study to investigate the role of digital imaging of biomarkers in case of melanoma"
2. Introduction needs proper modification. The current introduction is insufficient. Current text of introduction must be moved in results section as first section. Introduction needs completely new text paragraphs (in this order: background (1--2 paragraph)), Problem and complexity from previous known study (1--2 paragraph), The goal (1 short paragraph), and last section what you did and how? (1 paragraph)
3. Figure 2 and its following text must be moved to result section 2nd subsection
4. It would be worth to mention the resolution in Figure 2 and the same needs to be discussed in method section.
5. move figure 3 and the relevant text in result section
Author Response
The work titled as "The Role of Digital Imaging Biomarkers for Melanoma Detection Among Dermatologists: A reader study" was presented with the goal to better identify the biomarkers in Melanoma. The overall goal is promising but the manuscript needs to be improved. I have some comments which need to be implemented.
- Title must be updated so that it must look like research article and must represent the overall work presented here. My suggested title: "A deep learning based study to investigate the role of digital imaging of biomarkers in case of melanoma"
Comments: This is not a deep learning study and that's why it is novel. While this study does use machine learning, it is not "deep" in the sense of using hundreds of millions of features and combinations that will never be able to be processed by humans. Instead, is uses non-deep machine learning in a human visual language. We propose the title ‘The impact of melanoma imaging biomarker cues on detection sensitivity and specificity of clinically atypical nevi versus melanoma" The title has been updated in the main text.
Another option: ‘The role of imaging biomarker cues in enhancing sensitivity and specificity of melanoma detection versus atypical nevi’
- Introduction needs proper modification. The current introduction is insufficient. Current text of introduction must be moved in results section as first section. Introduction needs completely new text paragraphs (in this order: background (1--2 paragraph)), Problem and complexity from previous known study (1--2 paragraph), The goal (1 short paragraph), and last section what you did and how? (1 paragraph)
Comments: Per this request we have move the relevant paragraphs in the previous introduction to the results section. The first paragraph in the results section was modified slightly to improve flow. It was not clear if this reviewer would like Figure 1 and text moved to the results section as well but it was. Please see our complete modifications in the article text.
- Figure 2 and its following text must be moved to result section 2nd subsection
Comments: Figure 2 and the resultant text has been moved to the results section.
- It would be worth to mention the resolution in Figure 2 and the same needs to be discussed in method section.
Comments: The resolution of Figure 2 (now Figure 1) is improved and also noted in the manuscript.
- Move Figure 3 and the relevant text in result section
Comments: Figure 3 and relevant text have been moved to the results section.
Please note the Figure numbers changed to improve the flow of the Results section. Thank you.
Reviewer 2 Report
Comments and Suggestions for Authors
The authors presented a paper about “The Role of Digital Imaging Biomarkers for Melanoma Detection Among Dermatologists: A reader study”.
The topic is absolutely interesting and worth investigating.
The methodology followed by the authors is clear and well presented and the mean results obtained are in line with expected outcomes.
I found particularly interesting the single results of the participants to this study and therefore I would like to ask the authors to comment on some points in the discussion section:
1) Please provide a clearer definition of “Expert Dermoscopist”
2) I could not help noticing that the baseline sensitivity of an Expert Dermoscopist was the same as the performance of a Dermatology Resident (73.6%): could you comment on that?
3) In one case the support of the IBCs turned out to be detrimental with a -2.6% of sensitivity: again could provide a possible explanation?
4) Would you agree with the conclusion that the use of IBCs could be of particular interest to specific clinicians: for example Mohs surgeons seem to benefit much more on average than other colleagues.
Author Response
The authors presented a paper about “The Role of Digital Imaging Biomarkers for Melanoma Detection Among Dermatologists: A reader study”. The topic is absolutely interesting and worth investigating. The methodology followed by the authors is clear and well-presented and the mean results obtained are in line with expected outcomes. I found particularly interesting the single results of the participants to this study and therefore I would like to ask the authors to comment on some points in the discussion section:
1) Please provide a clearer definition of “Expert Dermoscopist”
Comments: An expert dermoscopist is a medical professional, typically a dermatologist, who has specialized training and experience in using dermoscopy. For our study, the three expert dermoscopists had at minimum of 3 years of experience practicing dermoscopy. The experts in this study have received dermoscopy training through dedicated courses and curriculums or are fellowship trained by leaders in the field. They are also heavily involved in dermoscopy teaching at academic institutions or national courses and conferences, and typically specialize in evaluation of pigmented lesions.
This comment is addressed in the methods and results section.
2) I could not help noticing that the baseline sensitivity of an Expert Dermoscopist was the same as the performance of a Dermatology Resident (73.6%): could you comment on that?
Comments: It's not significant with an N=1, but another observation would be that the resident didn't improve much but the expert dermoscopists did. This may be because their visual language (dedicated and lengthy training of their neural networks is more developed) is translated within the IBC visual language of the technology whereas the dermoscopy resident couldn't interact with the technology as well, due to decreased utilization of these specific neural networks. Additionally, the absence of clinical images and lack of information regarding typical nevus patterns for each individual and subsequent dermoscopic image, is objective information that was lacking from this study, and may prove challenging for experts to diagnose who rely more heavily on the complete clinical and dermoscopic picture.
This comment is addressed in the discussion of the manuscript.
3) In one case the support of the IBCs turned out to be detrimental with a -2.6% of sensitivity: again, could provide a possible explanation?
Comments: For this participant, it was an unlucky combination of correct and incorrect answers. We noticed that this participant developed response patterns based on the machine output. For example, an incorrect answer when the machine was correct made the participant overly rely on the machine. When this occurred, the participant then put too much trust in the machine for evaluation of subsequent images when the machine was incorrect. We realize that neither the humans or machines are perfect, and our conclusion is that there may be a form of cognitive manipulation with this type of human/machine interface. Therefore, in our opinion, it is vital for humans to realize the importance of clinical gestalt when utilizing machine learning technologies in clinical practice.
This comment is addressed in the discussion of the manuscript.
4) Would you agree with the conclusion that the use of IBCs could be of particular interest to specific clinicians: for example, Mohs surgeons seem to benefit much more on average than other colleagues.
Comments: Yes, in this case, Mohs surgeons have a strong background in dermatology but may lack dedicated training in dermoscopy. A strong dermatology background, essential for clinical recognition of malignant lesions, coupled with imaging biomarker cues, is ideal in this scenario. This approach would leverage both clinical intuition and dermoscopic image-based cues, which when utilized even in the absence of clinical images, appeared to aid these participants in distinguishing melanoma from atypical nevi in our study.
This comment is addressed in the discussion of the manuscript.
Reviewer 3 Report
Comments and Suggestions for Authors
1. There are too few references. The author should increase and review the literature to about 30.
2. Authors should compare their experimental results with those of other methods, especially lists or drawings.
Author Response
- There are too few references. The author should increase and review the literature to about 30.
Comment: The references have been updated to include at least 30. References are also updated with EndNote software and in the MDPI format as requested.
- Authors should compare their experimental results with those of other methods, especially lists or drawings.
Comment: Yes, this comparison has been addressed in the discussion section.
Round 2
Reviewer 3 Report
Comments and Suggestions for Authors
No comments.